

**Understanding the potential of climate teleconnections to project future groundwater**
**drought**
Rust, William.[a]; Holman, Ian.[a]; Bloomfield, John.[b]; Cuthbert, Mark.[c]; Corstanje, Ron.[d]
a Cranfield Water Science Institute (CWSI), Cranfield University, Bedford MK43 0AL
b British Geological Survey, Wallingford, OX10 8ED
c School of Earth and Ocean Sciences, Cardiff University, Park Place, Cardiff, CF10 3AT
d Centre for Environment and Agricultural Informatics, Cranfield University, Bedford MK43 0AL
**Abstract**
Predicting the next major drought is of paramount interest to water managers, globally.
Estimating the onset of groundwater drought is of particular importance, as groundwater
resources are often assumed to be more resilient when surface water resources begin to fail.
A potential source of long-term forecasting is offered by possible periodic controls on
groundwater level via teleconnections with oscillatory ocean-atmosphere systems. However,
relationships between large-scale climate systems and regional to local-scale rainfall, ET and
groundwater are often complex and non-linear so that the influence of long-term climate cycles
on groundwater drought remains poorly understood. Furthermore it is currently unknown
whether the absolute contribution of multi-annual climate variability to total groundwater
storage is significant. This study assesses the extent to which inter-annual variability in
groundwater can be used to indicate the timing of groundwater droughts in the UK. Continuous
wavelet transforms show how repeating teleconnection-driven 7-year and 16-32 year cycles
in the majority of groundwater sites from all the UK's major aquifers can systematically control
the recurrence of groundwater drought; and we provide evidence that these periodic modes
are driven by teleconnections. Wavelet reconstructions demonstrate that multi-annual
periodicities of the North Atlantic Oscillation, known to drive North Atlantic meteorology,
comprise up to 40% of the total groundwater storage variability. Furthermore, the majority of
UK recorded droughts in recent history coincide with a minima phase in the 7-year NAO-driven
cycles in groundwater level, allowing the estimation of future drought occurrences on a multi-
annual timescale. Long-range groundwater drought forecasts via climate teleconnections



present transformational opportunities to drought prediction and its management across the North Atlantic region.

## 1. Introduction

Inter-annual variability detected in hydrometeorological datasets has long been associated with systems of atmospheric-oceanic (climatic) oscillation, such as El Niño Southern Oscillation (ENSO) and the North Atlantic Oscillation (NAO). Such periodic teleconnection signals have been detected in rainfall (Luković et al. 2014), evapotranspiration (Tabari et al. 2014), air temperature (Faust et al. 2016), and river flow (Su et al. 2018; Dixon, et al. 2011); however these periodicities are often weak when compared to the finer-scaled (daily to seasonal) variability that is typical of hydrometeorological processes (Meinke et al. 2005). By contrast, groundwater systems are expected to be particularly susceptible to inter-annual teleconnection influence, given their sensitivity to long-term changes in rainfall and evapotranspiration (Bloomfield & Marchant 2013; Forootan et al. 2018; Van Loon 2015; Folland et al. 2015), and their ability to filter fine-scale variability in recharge signals (Dickinson et al. 2014; Velasco et al. 2015; Townley 1995). Consequently, recent studies have focused on the detection of long-term periodic cycles in groundwater levels in Europe (e.g. Holman et al. 2009; Holman et al. (2011); Folland et al. (2015); and Neves et al. (2019)), North America (e.g. Tremblay et al. (2011); Kuss & Gurdak (2014)) and globally (e.g. Wang et al. (2015); Lee & Zhang (2011)), and their relationships with climatic oscillations. An understanding of inter-annual perioidicity strength in groundwater level may provide an improvement in long-lead forecasting of hydrogeological extremes (Rust et al. 2018; Meinke et al. 2005; Kingston et al. 2006), in part, by enabling such cyclical behaviour to be projected into the future. This is particularly apparent of groundwater drought, which is known to result from multi-annual moisture deficits (Van Loon 2015; Van Loon et al. 2014; Peters et al. 2006). Therefore, it is critical to quantify the absolute strength of all periodicities within groundwater levels so that



the strength of inter-annual cycles, the influence of teleconnections, and their contribution
towards groundwater droughts can be understood.
Existing studies into groundwater teleconnections use quantitative methods to detect periodic
behaviour in groundwater datasets and often their relationship with time series of climate
indices (used to measure the strength and state of climate oscillations). Common quantitative
methods range from temporal correlation analysis (Knippertz et al. 2003; Szolgayova et al.
2014) to more complex periodicity detection and comparison. These latter methods include
Fourier transform, (Nakken, 1999, Pasquini et al. 2006), singular spectrum analysis (SSA)
(Kuss & Gurdak 2014; Neves et al. 2019) and wavelet transformations (Fritier et al. 2012;
Holman et al. 2011; Tremblay et al. 2011). The wavelet transform (WT) has been shown to be
particularly skilful at detecting inter-annual periodic behaviour in noisy hydrogeological
datasets; detecting the influence of the NAO, ENSO and Atlantic Multidecadal Oscillation
(AMO) on North American groundwater levels (Kuss & Gurdak 2014; Velasco et al. 2015), and
the NAO, East Atlantic pattern (EA) and Scandinavian pattern on European groundwater level
variability (Holman et al. 2011; Neves et al. 2019). However, in order to enhance inter-annual
periodicity detection, many studies have used data processing methods that remove or
supress variability at the higher end of the frequency spectrum (e.g. winter or annual averaging
or conversion of time series to cumulative departures from mean (Weber & Stewart 2004)).
Due to this data modification, it is currently unknown whether the absolute contribution of multi-
annual climate variability to total groundwater storage is significant. This limitation makes
assessment of systematic linkages between climatic oscillations and groundwater level
response problematic (Rust et al. 2018). As a result, the fundamental question of whether
inter-annual teleconnection cycles in groundwater level are sufficiently strong to influence
hydrogeological drought remains largely unanswered. Given the potential for improved long-
lead forecasting, quantification of inter-annual variability in groundwater level represents an
opportunity to support efficient infrastructure investment, systems of water trading (Rey et al.
2018) and robust planning for groundwater drought.



The aim of this paper is to assess the extent to which periodic behaviour in groundwater level
produced by teleconnections, may be used as an indicator for the timing of groundwater
droughts.  In doing so, this paper develops and applies an improved method to describe and
characterise the absolute strength of periodic behaviour in groundwater level and its drivers
(rainfall and evapotranspiration). This aim will be met by addressing the following research
objectives:
1. Characterise dominant intra- and inter-annual periodicities in groundwater level
records across a range of aquifer types
2. Quantify the absolute strength of these inter-annual periodic groundwater level
oscillations compared to the total variability in groundwater levels
3. Qualitatively assess evidence for the control of climate teleconnections on identified
inter-annual periods
4. Assess the extent to which the timing of the inter-annual periodic groundwater level
oscillations align with recorded groundwater droughts
These objectives will be implemented on UK hydrogeology records, given the considerable
coverage of recorded groundwater level data in time and across the country (Marsh &
Hannaford 2008) however the methodologies developed can be applied to any regions.

## 5.  Data and Methods

2.1.   Groundwater data
Groundwater level time series from 59 reference boreholes covering all of the major UK
aquifers, with record lengths of more than 20 years and data gaps no longer than 24 months,
have been assessed in the study. These recorded groundwater level hydrographs range from
21 to 181 years in length, with an average length of 53 years. The sites are part of the British
Geological Survey's Index Borehole network and, in addition to their data coverage, have been
chosen as they exhibit representative and naturalistic hydrographs with minimal impact from



abstractions. They cover a range of unconfined and confined consolidated aquifer types and
have been categorised into 5 main aquifer groups; Chalk, a limestone aquifer comprising of a
dual porosity system with localized areas where it exhibits confined characteristics; Limestone,
characterised by fast-responding fracture porosity; Oolite characterised by highly fractured
lithography with low intergranular permeability; Sandstone, comprised of sands silts and muds
with principle inter-granular flow but fracture flow where fractures persist; and Greensands,
characterised by intergranular flow with lateral fracture flow depending on depth and formation
(Marsh & Hannaford 2008).





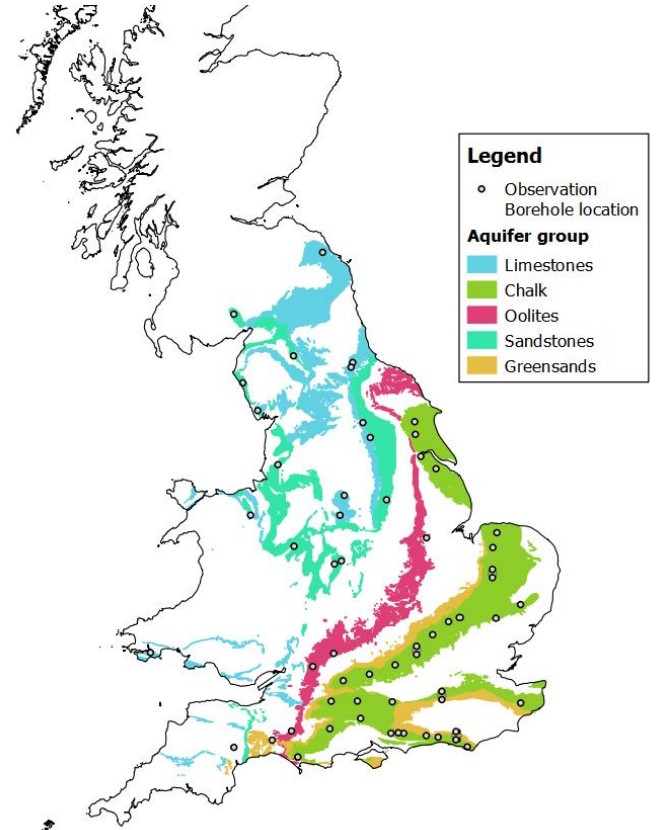


*Figure 1 - Location of the observation borehole locations used in this study. Boreholes within 0.5 km of another*
*have been displaced and denoted on a grey circle for visibility.*

**2.2. Rainfall**
Rainfall time series from the Centre of Ecology and Hydrology's CEH-GEAR 1km gridded
rainfall dataset (Tanguy et al. 2016), which is based on spatio-temporal interpolation of daily
rain gauge totals between 1890 and 2017, was used. However, relatively few rainfall stations
exist prior to 1950 that were used for this interpolation; as such data prior to 1950 was not
used in this analysis. Monthly rainfall series have been calculated for each borehole from the
1km grid cell in which they are located, as geospatial data on areas of groundwater recharge
connected to specific observation boreholes does not exist. This dataset may contain artefacts
as a result of the spatio-temporal interpolation, in comparison to station data. However the



use of rainfall data in this study is to provide a broad understanding of rainfall periodicities to
supplement those from groundwater level data. As such, this interpolated dataset is deemed
appropriate.

2.3.    Potential Evapotranspiration (PET)
Monthly PET series for each borehole have been derived from the Centre of Ecology and
Hydrology's CHESS-PE 1km gridded dataset of calculated daily PET values. The PET values,
between 1960 and 2015, were calculated using the Penman-Monteith equation, with
meteorological data taken from the CHESS gridded meteorological dataset. Details on the
underlying observation datasets and interpolation methods can be found in Robinson et al.
(2016). This data has been used previously to study long-term trends in hydrological variability
(Robinson et al. 2017).

2.4.    Methods
2.4.1.  Data pre-processing
In this study we use the continuous wavelet transform (CWT) to produce a time-averaged
frequency spectrum for each borehole hydrograph and co-located rainfall and PET time series.
For all datasets, gaps less than two years were infilled using a cubic spline to produce a
complete time series for the CWT. This interpolated information was later removed from the
time-frequency transformation (prior to time-averaging) to ensure that the data infilling had
minimal effect on the final spectrum. For time series with gaps greater than two years, the
shortest time period before or after the data gap was removed to produce one complete
record. Individual rainfall and PET time series were trimmed to match the length of the
corresponding borehole level time series. All time series were centred on the long-term mean
and normalized to the standard deviation to produce a time series of anomalies. Unlike most



previous studies, no high- or low-band filtering was undertaken on the datasets, ensuring all
information on periodic variability was preserved. This approach ensures that the Proportion
of a periodicity to the variance (standard deviation) of the original dataset is not modified.
2.4.2.  Continuous Wavelet Transform.
Following the data pre-processing steps, a CWT was applied to quantify the time-averaged
frequency spectra of the rainfall, PET and groundwater datasets. The CWT has been used
to assess long term trends and periodicities in many hydrological datasets including rainfall
(Rashid et al. 2015), river flow (Su et al. 2017), and groundwater (Holman et al. 2011; Kuss
& Gurdak 2014). We use the package "WaveletComp" produced by Rosch & Schmidbauer
(2018) for all transformations in this paper.
The continuous wavelet transform, $W$, consists of the convolution of the data sequence $(x_t)$
with scaled and shifted versions of a mother wavelet (daughter wavelets):

$$W(\tau, s) = \sum_t x_t \frac{1}{\sqrt{s}} \psi * \left( \frac{t - \tau}{s} \right)$$

(Eq. 1)

where the asterisk represents the complex conjugate, $\tau$ is the localized time index, $s$ is the
daughter wavelet scale and $dt$ is increment of time shifting of the daughter wavelet. The
choice of the set of scales $s$ determines the wavelet coverage of the series in its frequency
domain. The Morlet wavelet was favoured over other candidates due to its good definition in
the frequency domain and its similarity with the signal pattern of the environmental time
series used (Tremblay et al. 2011; Holman et al. 2011).
The CWT produces a time-frequency wavelet power spectrum for each time series.  Within
the time-frequency spectra, a cone of influence (COI) is used to denote those parts that are
affected by edge-effects, where estimations of spectral power are less accurate.  Therefore
only data from within COI were averaged over time to produce a time-average wavelet power
spectrum for frequency bands from 6 months up to 64 years. Wavelet power spectra were
then normalised to the maximum average wavelet value so that the frequency distribution of





each site can be directly compared. The normalized average wavelet power spectra (herein
referred to as the wavelet power spectra) provide a comparative measure of the strength of
the range of periodicities within frequency space.
2.4.3.  Significance testing
As Allen and Smith (1996) demonstrate, geophysical datasets can exhibit pseudo-periodic
behaviour as a result of their lag-1 autocorrelation (AR1) properties. Datasets with greater
AR1 tend to have spectra biased towards low frequencies, thus they are described as
containing red noise (Allen et al. 1996; Meinke et al. 2005; Velasco et al. 2015). In order to
assess the likelihood that a periodic signal is the result of internal (red) noise within the data,
significance of the red noise null hypothesis was tested. For this, 1000 randomly constructed
synthetic series with the same AR1 as the original time series were created using Monte Carlo
methods. Wavelet spectra maxima from these represent periodicity strength that can arise
from a purely red noise process. Wavelet powers from the original dataset that are greater
than these "red" periodicities are therefore considered to be driven by a process other than
red noise, thus rejecting the null hypothesis. Here, while a 95% Confidence Interval (CI) (<=
0.05 alpha values) is identified, we report on the full range of alpha results to provide a detailed
assessment of the likelihood of external forcing on periodic behaviour.
2.4.4.  Time reconstruction
In order to assess the characteristics of periodicities over time, we employ a reversal of the
wavelet transform (wavelet reconstruction) to convert selected periodic domains back into a
time series of normalised anomalies. Period bands were selected where the frequency spectra
identified shared wavelet power (and significance) between groundwater, rainfall and PET,
indicating a wide-spread signal presence at these bands.
The reverse wavelet transform is given by:


$$(x_t) = \frac{dj \cdot dt^{1/2}}{0.776 \cdot \psi(0)} \sum_s \frac{Re(W(.,s))}{s^{1/2}}$$

(Eq. 2)

Where *dj* is the frequency step and *dt* is the time step.
Negative phases of these time-reconstruction anomaly time series were compared to
episodes of recorded wide-scale hydrogeological drought (provided by Marsh et al. (2007) and
Todd et al. 2013)), to assess the relationships between inter-annual variability in groundwater
and groundwater droughts.
2.3.5 Periodicity strength quantification
While the wavelet power spectra from the CWT provide an estimate of the relative strength of
periodicities compared to the total frequency spectra, they do not provide an absolute measure
of a periodicities contribution to total groundwater variability (which includes noise and non-
periodic information). As such the percentage contributions of each time-reconstruction have
been calculated. Since the datasets were normalised to the standard deviation of the raw data
prior to the CWT, the standard deviations of the reconstructed anomaly time series represent
the proportion of the original standard deviation as a decimal percentage.





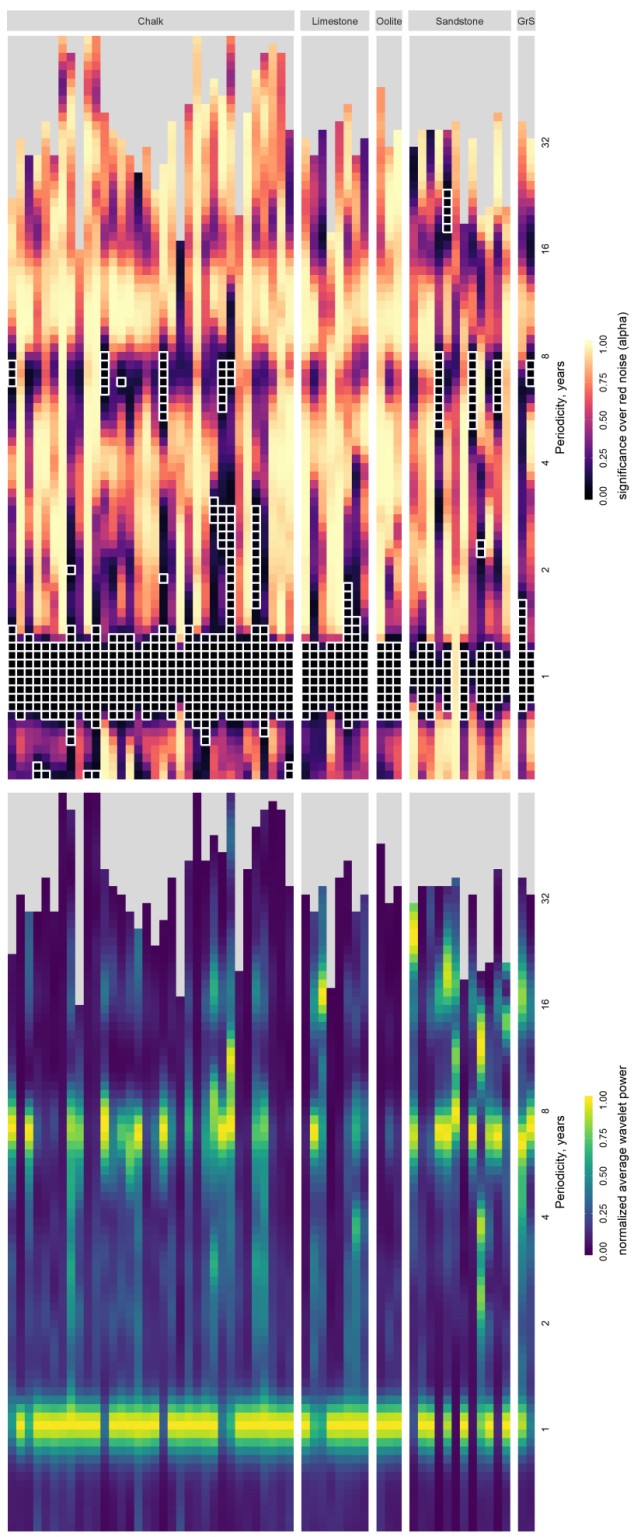

*Figure 2 - Normalised average wavelet power spectra (left) and wavelet power significance alphas (right) for monthly groundwater levels in the 59 index boreholes (grouped by aquifer type). In the right-hand figure, boxes outlined in white are those powers that are significant over red noise to a 95% confidence interval (a <= 0.05).*





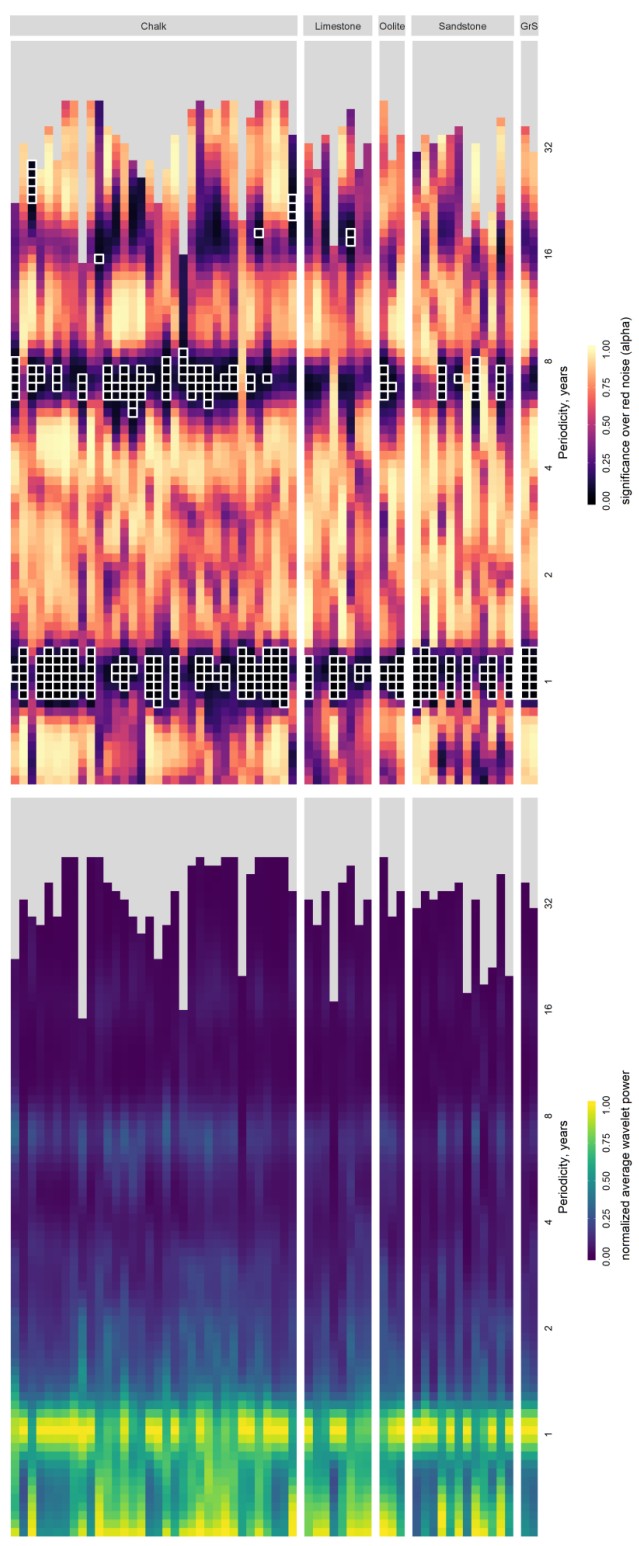

*Figure 3 - Normalised average wavelet power spectra (left) and wavelet power significance alphas (right) for monthly rainfall time series for co-locations of the 59 index boreholes.*

*In the right-hand figure, boxes outlined in white are those powers that are significant over red noise to a 95% confidence interval (α <= 0.05).*











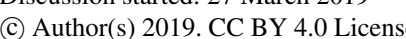

Figure 4 – Overlaid reconstructions of the three key periodic domains found across the 59 groundwater wavelet spectra are shown. All periods (both significant and non-significant)

within these bands have been displayed to allow for comparison of period strength and phase over time. Areas shaded blue represent periods of significant droughts in the UK.

Only reconstructions between 1955 and 2017 are shown to allow clearer comparison.







*Figure 5 – Overlaid rainfall (left) and PET (right) reconstructions of the three key periodic domains are shown. All periods (both significant and non-significant) within these bands have been displayed to allow for comparison of period strength and phase over time. Areas shaded blue represent periods of significant droughts in the UK. Only reconstructions between 1955 and 2017 are shown to allow clearer comparison.*






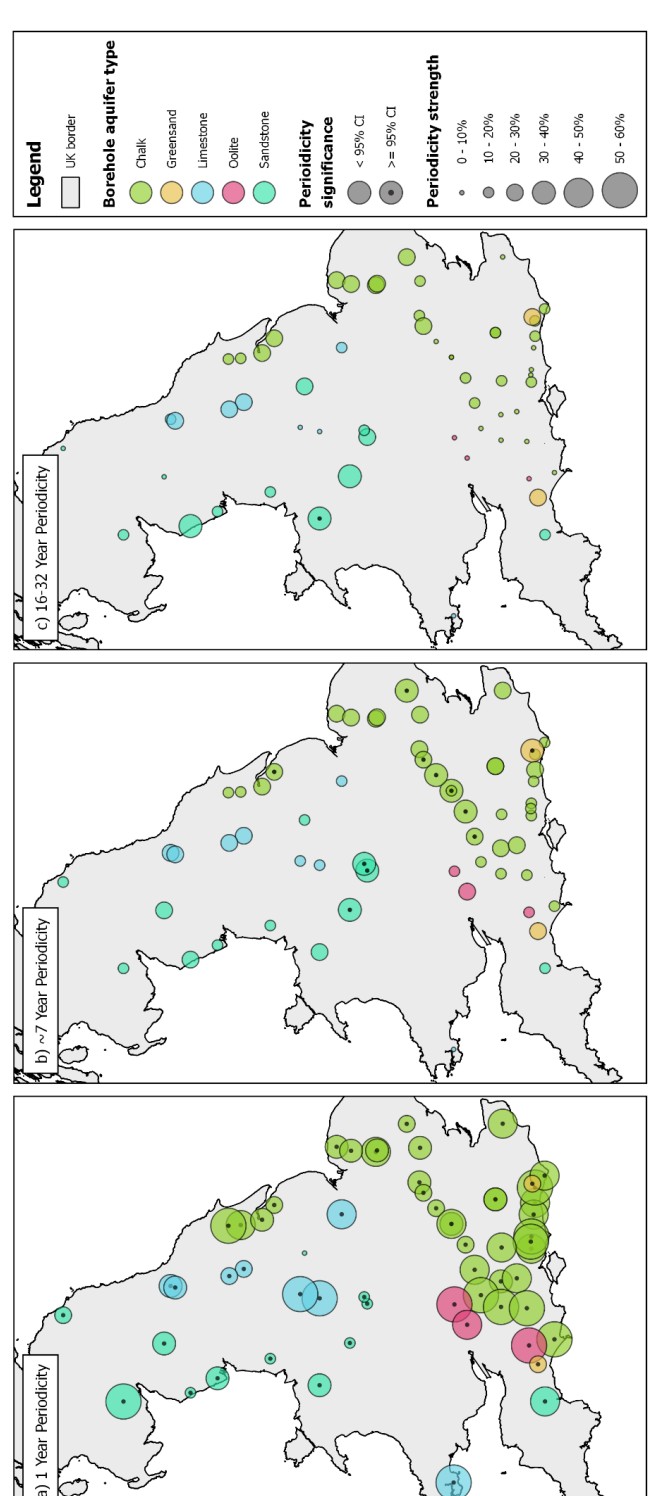

Figure 6 – Maps showing strength (percentage of the original time series standard deviation) and significance of the a) 1 year, b) ~7 year and c) 16-32 year periodicity bands.

No periodicity strength was found to be above 60% of the original signal.








## 3. Results

### 3.1. Time-averaged wavelet power and significance over red noise

Wavelet power spectra (frequency strength) and alpha values (significance) for each of the 59 groundwater level and rainfall time series are displayed in figures 2 and 3 respectively. Wavelet power is analogous to the strength of the periodicity compared to other frequencies. Periodicities with alpha values less than or equal to 0.05 (95% CI) are highlighted. Bands of greater wavelet power and lower alpha values at periodicities of 1, ~7 and 16-32 year(s) can be seen across the majority of the groundwater and rainfall spectra for the 59 sites (herein referred to as P1, P7 and P16-32 respectively). PET wavelet spectra were found to have no notable or significant periodicity beyond seasonality (indicative of the UK's temperate climate), and are displayed in the supplementary material.

The annual cycle (P1) exhibited the greatest power across 43 of the 59 observation borehole spectra, with normalised wavelet powers ranging from 0.03 to 1 (mean of 0.84). Alpha values for P1 in the observation boreholes also showed the greatest likelihood of external forcing when compared to the other identified periodic domains (alpha values ranging from 0.00 to 0.94, mean of 0.017). All but one observation borehole (site 51 showed significant (95%) alpha values for P1 wavelet power. Lower than average P1 wavelet powers were most prevalent in the Sandstone lithology (6 out of 12 sandstone sites), Greensands (1 out of 2 sites) and to some extent, the Chalk (6 out of 35 sites). P1 wavelet power was generally lower across all the corresponding rainfall time series, which is expected given rainfall's established bias towards high-frequency variability (Meinke et al. 2005). Of those boreholes with lower P1 power in groundwater, most (e.g. 35, 59) show greater P1 powers in rainfall (and PET) indicating hydrogeological processes as the mechanism for weaker P1 periodicity. However, a small number (e.g. 38, 40 and 42) had similarly low P1 periodicity in the corresponding rainfall, indicating meteorological drivers for poor annual strength at these observation





boreholes (considering that PET showed little variance in P1 strength across the observation
boreholes). PET spectra and alpha values showed a universally high P1 wavelet power.
The second greatest wavelet power across the groundwater boreholes was between 6 and 9
years, roughly centred on the 7 year periodicity (P7)). Maximum normalised groundwater
wavelet powers ranging from 0.01 to 1 (average of 0.52) between boreholes were detected,
and a corresponding band of lower than average alpha values (ranging from 0.01 to 0.99,
mean of 0.34), indicating that this periodicity is likely to be driven by an external variance.
Average P7 wavelet power values were greatest for Sandstone (0.68) and Greensands (1.00),
and lower for Limestone (0.39) and Oolite (0.17). Chalk showed intermediate strength with the
greatest range (0.01 to 1.00, mean of 0.50). Ten groundwater sites showed significant (95%)
P7 wavelet powers (sites 1, 12, 14, 19, 26, 27, 49, 53, 55 and 59). While the P7 wavelet power
in the corresponding rainfall data was considerably lower than those detected in groundwater
level (ranging from 0.014 to 0.35, mean of 0.16), the alpha values are comparable with the P7
signal strength in groundwater. This indicates that P7 signals in rainfall are weak, but likely
driven externally. Negligible wavelet powers and no significance was shown at the P7 band
for corresponding PET data.

The final and second mode of common inter-annual wavelet power was the band between 16
years and 32 years (P16-32). P16-32 had an average wavelet power of 0.28 across all
boreholes; ranging between 0.01 and 1. Similar to P7, the greatest wavelet power of P16-32
was found in the Sandstone (average of 0.58) and the Greensand (average of 0.64) aquifer
types. Whereas Chalk, Limestone and Oolite showed relatively weaker signals (averages of
0.18, 0.32 and 0.03 respectively). Only one site in the groundwater (site 50) and five rainfall
time series (sites 3, 11, 30, 34, 40) showed 95% significance over red noise in this periodicity
band.



### 3.2. Reconstructed anomaly time series

The three main common period domains identified by the wavelet transform (P1, ~7 and 16-32 years) were reconstructed into anomaly time series using the reversed wavelet transform and are presented in figure 4 for groundwater levels and figure 5 for rainfall and PET. This was undertaken to allow investigation and comparison of periodic behaviour over time and to assess how these reconstructed periodic signals, within multiple sites across multiple aquifers, align with periods of historical groundwater drought. The behaviour of the multiple reconstructed groundwater level, precipitation and PET anomaly time series (in all three periodicity domains) were shown to be well-aligned in time, with positive (maxima) and negative (minima) phases occurring within comparable time. The only exception to this pattern was seen between 1970 and 1980 in the P7 reconstructions, where phases in the P7 reconstructions become misaligned. This was predominantly apparent in groundwater and to a lesser extent in rainfall. Positive and negative phases of the P7 reconstructions in PET were well-aligned for the entire time series.

Notable episodes of groundwater droughts in the UK were overlaid onto the reconstructed periods in figure 5 between 1955 and 2016. With the exception of the 1975-6 event, every episode of drought in this time period coincides with a negative phase of the reconstructed P7 groundwater anomalies. The 1975-6 drought (often used as a benchmark drought in the UK due to its wide-reaching impacts (Marsh et al. 2007)) occurred at a time of notable minima/maxima misalignment of the P7 period in groundwater, and a period of negative anomaly in the P16-32 reconstructions. Most recorded major droughts in the UK appeared to occur irrespective of the state of the P16-32 anomaly, with droughts occurring in minima and maxima of this reconstruction.

### 3.3. Percentage standard deviation

The percentage of the standard deviation in the original groundwater level signal represented by each reconstructed periodicity band is shown in figure 4 for all the observation boreholes.





The percentages are representative of the absolute strength of the periodicity compared to
the recorded data variance (standard deviation).
P1 represents the greatest average contribution to groundwater variability across all the
aquifer groups (Chalk: 41%, Limestone: 40%, Oolite: 52%, Sandstone: 26%, Greensand:
28%). While most sites show that P1 accounts for the greatest proportion of the standard
deviation, P7 is the dominant periodicity at 11 of the 59 sites (5 within Sandstone, 5 within
Chalk and 1 within Greensand), and P16-32 is the strongest cycle in 3 of the 59 sites (3 within
Sandstone and 1 within Limestone). P1 strength in the Chalk appears to be greatest in the
South of England, with weaker strengths in the South East and East. Aside from the Chalk,
there are no clear spatial patterns in P1 strength. P7 accounts for an average of 21.7% of
signal strength across all aquifer groups, ranging from 3.8% to 40% across the observation
boreholes. Spatial variance in P7 signal strength is less when compared to P1, although there
is a noted area of significance in Chalk of South East England (e.g. the Chiltern Hills and
Cambridgeshire), and a smaller cluster of P7 significance in the Sandstone of the central
England, where the greatest P7 strengths are found. P16-32 strengths are spatially focused
in Eastern England for the Chalk, and the central and north-western England for the
Sandstone. No clear patterns for the remaining aquifer groups is apparent for the 16-32 year
periodicity band.



## 4. Discussion

The aim of this study was to assess the extent to which inter-annual cycles in groundwater levels (produced by teleconnections with climate oscillations) may be used to indicate the timing of future groundwater extremes. To achieve this, the absolute strengths of groundwater periodicities have been quantified and compared to the timing of historical droughts in the UK. In this wide-scale study, our results show for the first time that long-term cycles in groundwater levels are a crucial contributor to overall groundwater level variability. Additionally we show that much of this periodic behaviour closely aligns with episodes of historical groundwater drought over the past 60 years. These findings move beyond previous groundwater teleconnection research in the UK (Holman et al. 2011) and internationally (Kuss & Gurdak 2014; Neves et al. 2019) to provide a robust measure of the absolute contribution of inter-annual periodicities to groundwater levels fluctuations. In the following, we discuss the findings presented in this paper within the context of the research objectives and the implications for improved water resource management.

### 4.1. Characterisation of signal presence and strength in groundwater level

Many studies have focused on the role of seasonality in defining groundwater variability, and the onset and severity of groundwater drought (Jasechko et al. 2014; Hund et al. 2018; Mackay et al. 2015; Ferguson & Maxwell 2010). While we show that the annual cycle is an important component of groundwater response, it is often not representative of overall behaviour, accounting for (on average) less than half of total groundwater level variability. Conversely, we show that inter-annual periodicities form an unprecedented proportion of total groundwater variability; with 41% of sites (24 out of 59) exhibiting inter-annual periodicity strength that is comparable to (within 10%), or greater than, seasonality. It is expected that the strength of inter-annual cycles in groundwater level will vary according to signal strength in recharge drivers (e.g. rainfall and evapotranspiration) and hydrogeological processes that lag or attenuate long-term changes in these recharge signals (Van Loon 2013; Van Loon 2015; Townley 1995; Dickinson et al. 2014). These two processes may explain the local differences





in signal strength between sites in aquifer types and geographically across the UK, as
displayed in our results. For instance, pronounced inter-annual variability (significant 7 year
cycles and stronger 16-32 year cycles) in the Chalk sites is generally associated with
catchments of thicker unsaturated zones, larger interfluves or areas of weaker corresponding
seasonality in rainfall (for example, the Chiltern Hills in South East England). These catchment
properties have been shown to dampen higher frequency variability between rainfall and
groundwater response due to storage buffers, thereby producing a sensitivity to inter-annual
variability (Peters et al. 2006; Van Loon 2013). Inter-annual cycles are also generally strong
for the granular porosity aquifers (Sandstone and Greensand); which is to be expected given
the influence of lower hydraulic diffusivity (typical of granular porosity flow) on the suppression
of high-frequency variability (Townley 1995). This also agrees with Bloomfield & Marchant
(2013) who document sensitivity to long-term accumulation in rainfall in UK Sandstone
aquifers. Conversely, the Limestone and Oolite aquifer types exhibit weaker inter-annual
periodicities in groundwater level, with strong seasonality. Townley (1995) and Price et al.
(2005) document that, due to their faster-responding fracture porosity with low storativity,
limestone lithographies have a lower damping capacity of finer-scale variability in recharge,
meaning they are able to respond in-time to the strong seasonality in PET and rainfall. We
demonstrate that inter-annual periodicity strength in groundwater level is the result of both
meteorological (principally rainfall) and hydrogeological processes. It might appear that our
results show lower percentage contributions of inter-annual periodicities to total groundwater
level variability than those presented in other studies (Kuss & Gurdak 2014; Neves et al. 2019;
Velasco et al. 2015).  However as previous research has amplified lower frequencies in
groundwater level data spectra, the percentages reported in this study (which has not modified
the spectra of groundwater datasets prior to spectral decomposition (wavelet transform))
represent, for the first time, the absolute contribution of inter-annual variability to groundwater
level behaviour.





**4.2.    Evidence for teleconnection control on inter-annual groundwater variability**
Here, we discuss the evidence that the inter-annual variability present in UK groundwater level
records (as previously discussed) is the result of teleconnection influences with climatic
oscillations. The conceptualisation of groundwater teleconnections of Rust et al (2018)
suggests that a teleconnection between the oscillatory climate systems and groundwater level
would be associated with;
a) an apparent and coherent inter-annual periodicity band within groundwater sites

across a wide geographical area, that aligns with known inter-annual variability in

indices of climatic oscillations (for instance, the 7-year periodicity of the NAO (Hurrell

et al. 2003),

b) increased likelihood that this periodicity band is the result of an external influence, and

not the result of internal red-noise variability of the groundwater level time series (as

indicated by Allen et al. (1996) and Meinke et al. (2005))

c) comparable signals in rainfall as established drivers for inter-annual groundwater

variability, and

d) broad alignment of minima and maxima of time-reconstructed inter-annual

periodicities. Some fine-scale misalignment in groundwater periodicities is expected

as a result of unsaturated and saturated zone lags between rainfall and groundwater

response (Van Loon 2013; Peters et al. 2006; Dickinson et al. 2014; Cuthbert et al.

2019).

The majority of groundwater level hydrographs and corresponding rainfall profiles showed a
coherent band of increased periodicity strength and periodicity significance principally around
the 7-year frequency range and, to a lesser extent, the 16-32 year range. The 7 year periodicity
closely compares to the principle 7-year periodicity documented in the strength of the NAO's
atmospheric dipole, which has been associated with inter-annual periodicities in rainfall
(Meinke et al. 2005) and groundwater globally (Tremblay et al. 2011; Kuss & Gurdak 2014;
Holman et al. 2011; Neves et al. 2019). Additionally, the time-reconstructions show clear





temporal alignment of minima (with the exception of the 1975-6 period, which will be discussed
later), indicating the wide-spread coherent influence of a climatic teleconnection. As such, we
corroborate with existing research that documents the control of the NAO on UK rainfall
(Alexander et al. 2005; Trigo et al. 2004), and show new evidence of the wide-spread
propagation of inter-annual variability in rainfall through to spatio-temporal inter-annual
groundwater variability, conceptualised by Rust et al (2018).
While the NAO is known to be the dominant mode of winter climate variability in Europe
(López-Moreno et al. 2011; Alexander et al. 2005; Hurrell & Deser 2010), the second strongest
is provided by the East Atlantic (EA) pattern (Wallace & Gutzler 1981). The EA is similar in
frequency structure to the NAO but shifted southward, however it has been shown to exhibit
its own internal variability (Hauser et al. 2015; Tošić et al. 2016; (Moore et al., 2013).
Importantly, the EA has been shown to exhibit a 16-32 year periodicity (Holman et al, 2011),
and therefore aligns with the second strongest mode of inter-annual variability in groundwater
and rainfall documented in this study. Similar to the 7-year periodicity, the 16-32 year cycle
detected in groundwater levels shows an increased likelihood of external variance, and
temporal alignment of minima and maxima when reconstructed back into the time domain. As
such, we consider the EA to be the ultimate driver of the 16-32 year periodicity detected in UK
groundwater level. While the EA has received little focus in climate variability research
compared to the NAO, our findings here align with Krichak & Alpert (2005) who document a
multi-decadal control on UK and European precipitation through shifting phases of the EA,
and Holman et al (2011) who detected weak relationships between the EA and groundwater
levels in the UK. Comas-Brua and McDermotta (2014) suggest that much of the multi-decadal
climate variability (temperature and precipitation) in the North Atlantic region can be explained
by a modulation of the NAO by the EA, which may contribute to the spatial and temporal
variability seen in both the ~7 year and 16-32 year reconstructions across the borehole sites.
In summary, we assert that the inter-annual variability detected in UK groundwater and rainfall
data is likely the result of a climatic teleconnection with both the NAO and the EA. As such,



we document the first evidence of the absolute strength of both the NAO and the EA's control
on 7-year and 16-32 year varaibility in groundwater systems respecitvely.
**4.3.    Teleconnections as indicators for groundwater extremes**
The final objective of this paper was to assess the extent to which the timing of inter-annual
periodic groundwater level oscillations align with the timing of recorded groundwater droughts.
To achieve this, documented periods of groundwater drought have been compared to
reconstructed periodicities within groundwater level. We show that every documented
groundwater drought between 1955 and 2014 aligns with a negative phase of the ~7 year
cycle detected in the majority of UK groundwater boreholes, with the exception of the 1975-6
drought. In addition to the strength of a ~7 year cycle in UK groundwater level previously
discussed, this alignment provides strong evidence that the NAO influences inter-annual
groundwater variability, resulting in groundwater drought on an approximate 7-year
recurrence. As mentioned, the only drought that does not fit this pattern is the 1975-6 drought,
which occurred during the only episode of temporal misalignment in the reconstructed 7-year
groundwater level periodicities. The 1975-6 drought is of particular interest for the UK as it is
often used as a benchmark drought, being one of the most severe droughts in recent history
(Marsh et al. 2007). (Rodda & Marsh 2011) attributed the severity of this drought to several
short-term influences (such as positive pressure anomalies driving dryer conditions), in
contrast to the multi-year accumulation of moisture deficits that typically result in
hydrogeological drought, particularly in the UK (Van Loon 2015; Bloomfield & Marchant 2013).
These assertions are apparent in our results, as the time reconstructions show negligible multi-
annual groundwater response occurring during this period, and an intense short-term
suppression of seasonality. As such, these results infer that the NAO did not directly modulate
the 1975-6 drought, agreeing with Parry et al (2011) who found no relationship with this
drought and the NAO. This potentially points to both the 1975-6 drought and the disrupted
NAO being modulated in parallel by a wider atmospheric control. Peings & Magnusdottir
(2014) suggest that atmospheric blocking prohibits the expected effects of the NAO on UK





and European hydrology, which may indirectly explain both the 1975-6 drought and the
disruption to the 7 year periodicity in UK groundwater (Rodda & Marsh 2011).
While the 16-32 year periodicity in groundwater level does, in general, align with historical
recorded droughts, this is not as coherent as with the 7 year cycle, with droughts occurring in
positive and negative phases. However, given the percentage contributions of the 16-32 year
periodicity to total groundwater variability, it is likely that this signal has a role in modulating
the severity of droughts influenced by the NAO 7-year drought cycle (as suggested by Comas-
Brua and McDermotta (2014)). For instance, the severe 1975-6 drought occurred at a slight
negative phase of the 16-32 year cycle, indicating that a portion of the severity of this drought
was the result of EA's influence.

Based on the alignment of the 7 year cycle (and partial alignment of the 16-32 year cycle) with
historical recorded UK droughts, we conclude that the NAO (and EA) directly modulate the
severity and timing of droughts in the UK. Furthermore, the 7-year cycle is shown to be a
sufficient indicator of the onset of droughts, based on this historical alignment. Consequently,
this 7-year cycle can be extrapolated beyond the end of the dataset used in this study.
Therefore, based on a projected 7-year cycle, we predict the UK will likely enter drought
conditions around 2018/19, 2025/6 and 2033/4, assuming the continuation of the NAO
system's influence. This projection is further validated by the onset of drought conditions in
the UK in mid-2018 (Hannaford, 2018).

In the UK, the economic regulator has implemented several measures to promote the trading
of water between water supply companies to enable a more robust water supply system
(OfWAT 2019; Deloitte LLP 2015). Here, we show that recursive patterns in groundwater
contribute to a considerable proportion of the total groundwater level variability and therefore
may provide new insights to allow undertakers of water supply to trade water further into the





future, depending on teleconnection sensitivities. Such forecasted planning could help to
reduce the ecological and human impacts of groundwater drought by allowing more time to
plan and organised the required water transfers from areas less susceptible to teleconnection-
driven drought.

**5.  Conclusions**
This paper assesses the role of inter-annual variability and ocean-atmosphere systems in
influencing groundwater drought. We quantify, for the first time globally, the absolute
contribution of inter-annual cycles to groundwater variability, and provide new evidence for the
influence of the NAO's control of European rainfall on UK groundwater drought over the past
60 years.
The wavelet transformation was used to identify and evaluate bands of periodic external
influence on UK groundwater level hydrographs, documenting the strength of a 1, approximate
7 and 16-32 year cycle in the majority of sites assessed. We find that seasonality accounts for
an average of 39% of groundwater level variance across boreholes; with 7-year cycle
accounting for an average of 21%, and 16-32 years accounting for 15%. Furthermore, the
minima of NAO-driven cycles in groundwater level align with the occurrence of recorded
groundwater drought, allowing the estimation of future drought occurrences on a multi-annual
timescale. The analysis demonstrates that the NAO is the principle control (and the EA as the
secondary control) on inter-annual variability in UK groundwater level, and provides a new
approach to forecast the onset of groundwater droughts through an extrapolation of cyclical
behaviour into the future.  As such we identify 2018/19, 2025/6 and 2033/4 as likely episodes
of future droughts in the UK. Although further work is required to better understand the
teleconnection sensitivity, the methods described in this paper provide a robust and
transferable approach for assessing the quantitative influence of teleconnections in
hydrological datasets.  It is clear from our results that long-range groundwater drought



forecasts via climate teleconnections present transformational opportunities to drought
prediction and its management across the North Atlantic region.

**Acknowledgements**
This work was supported by the Natural Environment Research Council [grant number
NE/M009009/1], and the British Geological Survey (Natural Environment Research Council).
We acknowledge the British Geological Survey for provision of the groundwater level data,
and the Centre for Ecology and Hydrology for provision of the CHESS rainfall data
(https://doi.org/10.5285/33604ea0-c238-4488-813d-0ad9ab7c51ca) and CHESS PET data
(https://doi.org/10.5285/8baf805d-39ce-4dac-b224-c926ada353b7).     John     Bloomfield
publishes with the permission of the Executive Director, British Geological Survey (NERC).
Mark Cuthbert acknowledges support for an Independent Research Fellowship from the UK
Natural Environment Research Council (NE/P017819/1). We thank Angi Rosch and Harald
Schmidbauer for making their wavelet package "WaveletComp" freely available.
The groundwater level data used in the study are from the WellMaster Database in the
National Groundwater Level Archive of the British Geological Survey. The data are available
under     license     from     the     British     Geological     Survey     at     https:
//www.bgs.ac.uk/products/hydrogeology/WellMaster.html (last access: 26/03/2019).

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
