# Peer review of "Understanding the potential of climate teleconnections to project future groundwater"

_Hydrology and Earth System Sciences, 2019_

## Referee Comment (RC1) · Anonymous Referee #1 · 13 Apr 2019

The paper analyses cyclic variations in groundwater levels across 59 boreholes in the UK, classified according to aquifer type. Cycles of 1, ∼7 and 16-32 years were recognized using the wavelet transform method. The most significant oscillatory component is the ∼7-year cycle, which is driven by the North Atlantic Oscillation. The minimum phases of this component coincide with some of the major droughts in the UK. The connection between groundwater level/rainfall variability and both the teleconnections and droughts is qualitative but based on substantial evidence from published literature. The topic of groundwater drought forecasting is vital in the actual context of climate change, and the paper is adequately structured and well written. However, it has several shortcomings that need correction, namely:

1. It is not correct to say that this paper quantifies the teleconnections contribution to

the absolute groundwater variability for the first time (line 346, 392, 449, 509). The authors claim that all previous studies performed low-pass filtering or some averaging of groundwater level time-series before wavelet transform or PCA methods. This is not so, at least in the case of Tremblay et al., 2011 and Neves et al., 2019.

2. The proportion of groundwater variability driven by teleconnections in the UK seems indeed much lower than in other parts of the world. Blaming the amplification of low-frequencies in other studies (that does not happen) is therefore not valid, and the authors should seek other explanations. The results may probably be a consequence of the specific climate and hydrogeologic conditions in the UK, but may also be a consequence of the different methodology used to compute the percentages of variance. Do the authors get the same results using SSA or PCA? One alternative method should be used in order to be sure.

3. A closer look at Figure 4 shows time intervals between droughts of approximately 2.5, 3, 5, 6 and seven years. Therefore, it seems excessive to declare that the approach presented in this paper can be used to predict droughts with a recurrence of seven years (line 492). Moreover, the authors do not even mention the non-stationarity of teleconnections and ignore the effects of global warming on the predictability and statistics of extreme events. The authors need to re-write some parts of the text and elaborate more on these issues.

Minor points: - Please increase the font size of text and labels in the pictures - Line 283: can you explain better why the 7-year cycle has greater significance values in rainfall than in groundwater? - Line 315: do you mean misalignments amongst borehole records? Are there consistent misalignments amongst aquifers? - Line 321: figure 6 instead of figure 4? - Lines 342-354: the whole paragraph is redundant and would better be omitted.

---

## Referee Comment (RC3) · Anonymous Referee #2 · 13 May 2019

The manuscript links inter-annual variations in groundwater levels across the UK a to large scale climate influence. I appreciated the well structured, data-driven approach and mainly agree with the well discussed findings. Nevertheless, some claims that the authors try to express are formulated stronger than supported by the presented results. The manuscript is generally well written and a significant new contribution to understanding climate influence on groundwater droughts. In my opinion, it would make a valuable contribution to HESS after implementing some changes and addressing the following points:

Major comments: - In general the interpretation of trends by aquifer type is tricky for Oolite and Greensand sites as there are only 2 and 3 observation boreholes. I recommend clearly stating the number of observation boreholes in the introduction (somewhere in

the introduction between line 110 and 117) and afterwards avoiding (over)interpretation of statistic measures in these two aquifer types (e.g. lines 262, 277-278, 290-292, 325-326 . . .). Furthermore there is no strong differences between the aquifer types, at least I don't see these e.g. in Figure 6, in my opinion these differences are not shown in your results (line 365 – 369). Consider rephrasing to make a less strong claim. - The drought events used for comparison, do not occur in the 7-year cycles that are proposed for potentially predicting groundwater droughts in the UK. These drought events occur in different time intervals. To support teleconnection influences of larger scale climate phenomena you need to further elaborate on this. The claims in the discussion on the relation of NAO and EA to the 7 year and 16-32 year cycles of droughts are very strong considering the results; consider reformulating it - Key for the interpretation of section 3.2 is additional information on the drought periods you are referring to (green bands in Figures 4&5). It would be helpful to provide some background on these events (on magnitude and durations), this potentially also helps to improve the discussion on climatic teleconnections. - The discussion can be (and should be) considerably shortened by removing the first, very general and summarizing paragraph, also the last parts of the discussion are a little more messy than the rest of the manuscript, please consider re-organizing the discussion a little bit (see also minor comments) - In my opinion, the quality of the Figures is not sufficient for publication: please change size of labels, axis labels, legends e.g. in Figures 2, 3, 4 and 5. Add a scale bar to all GB maps (Figure 1, Figure 6 and sup. Figure 1) - Also in the conclusions we find some very strong statements that are in my opinion only partially supplied by your results: line 509 "we quantify, for the first time globally" (as pointed out before this is not the first time, see interactive comments); line 517 – 523 ". . . allowing the estimation of future drought. . ." (I would suggest changing this very strong claim accordingly, you show potential control of NAO and EA on groundwater droughts in the UK); line 527-529 "it is clear from our results . . . drought prediction and its management across the North Atlantic region" (inn my opinion you cannot say that from your results, you mostly qualitatively analyze the coinciding timing of drought and climate across the UK); I'd skip

line 524 – 527 at it is not very informative;

Minor comments: line 102 formatting error, change 5 to 2 line 104 – 106 how many boreholes did you consider for your analysis in the end? There is 59 in total, according to supplementary. In Table 1, 9 have gaps longer than 24 months? Please be more explicit. line 120 – 121 consider rewording caption of Figure 1: "Location of the observation borehole locations..." "denoted on a grey cycle" not clear, change to "denoted on one grey cycle" line 124 – 128 consider rewording: why are you mentioning data from prior 1950 if you only use data from 1950 onwards? line 152 "data gaps of greater than two years" you do not report in which time series these data gaps exist, previously you mentioned that gw data had "data gaps no longer than 24 moths" (line 105) line 149 – 152 is that a well-established thing to do? Can we read about this somewhere? line 165 mention that this is a package for R line172-173 "good definition in the frequency domain" not clear, please be more explicit line 230 / Figure 4: y-axis "PET residuals" line 265 – 271 there are indeed distinct differences in the driving forces of periodicity, maybe you can be more explicit in describing these? line 283 – 284 this is a very strong statement, can be more explicit? What do you mean by "likely driven externally"? line 313 – 316 long sentence, unclear, consider rephrasing, splitting line 321 Figure 6 instead of Figure 4 lines 335-338 consider rewriting to "in the sandstone aquifers of central England", and so forth line 360 refer to Figure 6 line 365 – 369 I don't see the "strong" differences between the different "hydrogeological processes" in the presented data (see also major comments) line 428 – 448 this paragraph is a little messy, consider re-writing (NAO and EA control climate variability across Europe, NAO by ... EA by...; what is their temporal resolution; how are they linked; how does that refer to your findings;) remove "(" in line 432; line 448 – 450 this is a very strong claim, you might want to reformulate this line 458 – 463 consider removing this sentences as they are repetitive; line 456 Rodda & Marsh (2011) line 468 "Van Loon, 2015" is a review paper, it does not talk about UK droughts particularly, throughout the manuscript this reference is used very often but it is not always suitable; line 465 – 477 this is not very well structured, see comments above, consider rephrasing the exceptionality of the 1975-

76 drought line 487 – 495 I would remove these two paragraphs, again very strong claims (especially regarding the predicted future drought dates); there might be larger uncertainties than expected, considering non linearity, changing climatic conditions,... line 516-517 the 7-year cycle accounted... and the 16-32 years cycle... (also: consider using past tense in this sentence and the next one) line 520 (and the EA is the secondary control) There are minor inconsistencies in the wording you use: e.g. mean (line 257, 282) vs. average (line 274, 291), wavelet power vs. wavelet strength; There are typos in: line 158 Proportion; line 203 indicate; line 307 became; There are many commas missing throughout the manuscript e.g. in: line 100 , however; line 114 sands, silts and muds; line 192 , that; line 214 , the percentage; lines 247 & 252 , respectively; line 307 ,and to; Please double check and be more careful! Please also double-check the References section for inconsistencies: e.g. line 577, line 585 (webpage?); line 611;

---

## Author Comment (AC2) · 19 Jun 2019

This appear to be a duplicate of the previous comment "Anonymous Referee #1, 13 Apr 2019".

---

## Author Response (AR1)

We would like to thank Anonymous Referee #1 for their detailed review comments. We 2 found them to be insightful, and, through our responses to them set out below, we believe that they have resulted in a much improved paper.

Major Comment 1: Referee #1 states that "it is not correct to say that this paper quantifies 5 the teleconnection contribution to the absolute groundwater variability for the first time (line 346, 392, 449, 509). The authors claim that all previous studies performed low-pass filtering 6 or some averaging of groundwater level time-series before wavelet transform or PCA 7 8 methods. This is not so, at least in the case of Tremblay et al., 2011 and Neves et al., 2019. The proportion of groundwater variability driven by teleconnections in the UK seems indeed q much lower than in other parts of the world. Blaming the amplification of low frequencies in 10 11 other studies (that does not happen) is therefore not valid, and the authors should seek other 12 explanations." 13 Response to Major Comment 1: The Reviewer is correct that there may be many other contributing explanations in some cases which we have now outlined in Lines 375 - 386. 14 15 However, we do also think that previous studies that have sought to quantify the proportion 16 of extra-annual cyclical variability in groundwater level and that may have used preprocessing steps that might have altered the strength of extra-annual periodicities within the 17 groundwater spectra. A key example is cumulative departure from the mean (CDM) which 18 19 has been undertaken by Neves et al., 2019. While not explicitly designed as a low-pass filter, 20 CDM is a process that amplifies low frequency periodicity and suppress higher frequency periodicities. This is, for example, exemplified in figure 4 in Neves et al., 2019 where we can 21 22 see little annual variability in rainfall: which we would not expect from a 'raw' dataset. As a 23 result of this, the strength of extraannual periodicities may be misrepresented when compared to the raw groundwater level data. Another example is given in Tremblay et al., 24 25 2011., while no preprocessing of the data is apparent, periodicities reported have not 26 included the strength of seasonality. As such, we cannot tell the actual strength (and 27 therefore importance) of the extra-annual periods, as we cannot tell how they compare to 28 seasonality (known to be a major component of hydrological processes). As such we believe 29 this paper provides an explicit assessment of the percentage of cyclical variability to the unaltered groundwater level data spectrum. We have amended the text in the locations 30 31 highlighted by Referee #1 to make this clearer, e.g. Lines 375 - 386, 495 - 502.

Major Comment 2: Referee #1 states that "The results may probably be a consequence of the specific climate and hydrogeologic conditions in the UK, but may also be a consequence of the different methodology used to compute the percentages of variance. Do the authors get the same results using SSA or PCA? One alternative method should be used in order to be sure."

Response to Major Comment 2: We appreciate why Reviewer #1 has made this observation and suggests additional analyses. However, SSA/PCA (which the co-authors applied to 38 39 groundwater level observations in Holman et al.(2009)) requires removal of trends (non-40 stationary) before any meaningful information on principal components can be extracted and 41 therefore implies stationarity. In addition, the aim of this paper was to identify specific 42 periodicity bands that are shared between groundwater hydrographs, and with SSA/PCA 43 there is no guarantee that eigenvectors between datasets will be comparable or even 44 periodicities of these can be confidently estimated (as one would have to again assume 45 stationarity to identify frequencies from principal components). Nevertheless, we have 46 extended our literature review to include potential other sources for these signal strengths in 47 light of the Reviewer's comment. E.g. Lines 479 – 492, 375 – 386

Major Comment 3.1: Referee #1 states that "A closer look at Figure 4 shows time intervals

- 49 between droughts of approximately 2.5, 3, 5, 6 and seven years. Therefore, it seems
- 50 excessive to declare that the approach presented in this paper can be used to predict
- 51 droughts with a recurrence of seven years (line 492)."
- Response to Major Comment 3.1: We agree with the comment that the wording around the recurrence of drought events is too strong and does not account for the variability in the time intervals between recorded droughts. To address this concern, we have now added a further review of drought mechanisms and have updated the text to refer to reflect drought risk,
- rather than the definite timings of drought in Lines 450 492. In addition, Figure 4 has been
   modified to better illustrate the drought start/end dates, although there is inevitable spatial
   uncertainty in these.
- 59 Major Comment 3.2: Referee #1 states "Moreover, the authors do not even mention the nonstationarity of teleconnections and ignore the effects of global warming on the predictability and statistics of extreme events. The authors need to elaborate more on these issues."
- and statistics of extreme events. The authors need to elaborate more on these issues.

Response to Major Comment 3.2: We agree with the Reviewer that more elaboration is needed on these issues, although we also note that the effects of global warming on the predictability and statistics of extreme events is a very broad and still developing subject. It is mentioned in the text that the varying strength (and therefore the non-stationarity) of the

NAO does not directly appear to influence the occurrence of historical drought, therefore wide-spread droughts appear sensitive to the NAO phase, rather than its overall strength.

However, we have now also added additional text to clarify these issues in Lines 479 – 492.

Minor Comment 1: Please increase the font size of text and labels in the pictures – Figures
 have been updated

Minor Comment 2: Line 283: can you explain better why the 7-year cycle has greater
 significance values in rainfall than in groundwater? *Text has been updated in Lines 287-289*

- Minor Comment 3: Line 315: do you mean misalignments amongst borehole records? Are
   there consistent misalignments amongst aquifers? *Text has been updated at line 319*
- 75 Minor Comment 4: Line 321: figure 6 instead of figure 4? Text has been updated

Minor Comment 5: Lines 342-354: the whole paragraph is redundant and would better be 77 omitted. *We agree that this paragraph is not required and have removed the text*

**79 Response to Reviewer #2 comments**

We would like to thank Anonymous Referee #2 for their detailed review comments. We found them to be insightful, and, through our responses to them set out below, we believe that they have resulted in a much improved paper.

Major comment 1: - In general the interpretation of trends by aquifer type is tricky for Oolite and Greensand sites as there are only 2 and 3 observation boreholes. I recommend clearly stating the number of observation boreholes in the introduction (somewhere the introduction between line 110 and 117) and afterwards avoiding (over)interpretation of statistic measures in these two aquifer types (e.g. lines 262, 277-278, 290-292, 325- 326 . . .). Furthermore there is no strong differences between the aquifer types, at least I don't see these e.g. in

Figure 6, in my opinion these differences are not shown in your results (line 365 – 369).

Consider rephrasing to make a less strong claim.

Response to Major comment 1: We agree that it is difficult to interpret patterns in response as a function of aquifer type, particularly for the Oolites and Greensands where there are only a couple of observations from each aquifer; that we should avoid over interpreting any
 of the aquifer specific results. Consequently, we have revised the text at L112-116 to explicitly state how many observations there are for each aquifer, and have added cautionary statements in the appropriate sections of text noting the relatively small sample sizes and the consequent difficulties in unambiguously identifying systematic differences in responses between the different aquifers, e.g. Lines 265 – 267, and we have avoided group-99 specific interpretation in the discussion for these groups.

Major comment 2: The drought events used for comparison, do not occur in the 7-year
 cycles that are proposed for potentially predicting groundwater droughts in the UK. These
 drought events occur in different time intervals

Response to Major comment 2: We agree that the wording around the recurrence of drought events was too strong and did not account for the different time intervals between recorded droughts. In response we have now included a further review of drought mechanisms and have updated the text to refer to reflect drought risk, e.g. Lines 469-566, rather than the definite timings of drought. In addition, Figure 4 has been modified to better illustrate the drought start/end dates, although there is inevitable spatial uncertainty in these.

Major comment 3: To support teleconnection influences of larger scale climate phenomena 100 you need to further elaborate on this. The claims in the discussion on the relation of NAO

and EA to the 7 year and 16-32 year cycles of droughts are very strong considering the results; consider reformulating it

Response to Maior comment 3: We have now softened our claims regarding the NAO and

EA control on groundwater and rainfall in the discussion, and included further literature review about the potential causes for these signals, see Lines 427 – 432 and 461- 463, and

*have removed lines* 464 – 467.

Major comment 4: Key for the interpretation of section 3.2 is additional information on the 118 drought periods you are referring to (green bands in Figures 4&5). It would be helpful to 119 provide some background on these events (on magnitude and durations), this potentially 120 also helps to improve the discussion on climatic teleconnections.

Response to Major comment 4: We have now included additional information on the drought 122 periods in the Discussion at Lines 481-557

Major comment 5: The discussion can be (and should be) considerably shortened by removing the first, very general and summarizing paragraph, also the last parts of the discussion are a little more messy than the rest of the manuscript, please consider reorganizing the discussion a little bit (see also minor comments)

Response to Major comment 5: We agree that this paragraph is not required and have
removed the text at Lines 347 - 359, and have reworded the final paragraph at Lines 576 –
and 581 - 605.

Major comment 6: In my opinion, the quality of the Figures is not sufficient for publication:
please change size of labels, axis labels, legends e.g. in Figures 2, 3, 4 and 5. Add a scale
bar to all GB maps (Figure 1, Figure 6 and sup. Figure 1).

Response to Major comment 6: Figures have been updated to include the suggested134 changes

Major comment 7: Also in the conclusions we find some very strong statements that are in my opinion only partially supplied by your results: line 509 "we quantify, for the first time 136 137 globally" (as pointed out before this is not the first time, see interactive comments); line 517 138 - 523 "... allowing the estimation of future drought..." (I would suggest changing this very strong claim accordingly, you show potential control of NAO and EA on groundwater 139 140 droughts in the UK); line 527-529 "it is clear from our results . . . drought prediction and its 141 management across the North Atlantic region" (inn my opinion you cannot say that from your results, you mostly qualitatively analyse the coinciding timing of drought and climate across 142 143 the UK); I'd skip Interactive comment line 524 - 527 at it is not very informative; 144 Response to Major comment 7: We have amended the text throughout the document to 145 focus more on the contribution to the existing knowledge base rather than claiming anywhere to be the first study to produce such findings. See, for example, Lines 495 – 516. 146 147 148 Amendments made to the manuscript 149 1. All instances of "extra-annual" have been amended to "multi-annual" for clarity 150 Information on sample sizes has been included in the description of datasets at Lines 2. 151 112, 114, 115 and 116 in response to Reviewer 2, Major comment 1 152 3. Scale bar has been added to Figure 1 at line 121 in response to Reviewer 2 Major comment 6 153 154 4. Text size increased in Figure 2, 3, 4 and 5, and scalebar added to figure 6 in 155 response to Reviewer 2 Major comment 6 5. Text added to caution against over interpretation of datasets with small sample sizes 156 157 at Lines 265-267 in response to Reviewer 2. Major comment 1 158 6. Explanatory text added to Lines 288 - 289 in response to Reviewer 1 Minor comment 159 2 160 7. Lines 319-320 have been amended in response to Reviewer 1 Minor comment 3 161 Lines 347 – 359 have been removed in response to Reviewer 1 Minor comment 5 8. and Reviewer 2 Major comment 5. 162 163 Lines 390 - 403 have been amended according to Reviewer 1 Major comment 1 and 9 164 2 and Reviewer 2 Major comment 3 165 10. Lines 446 – 453 have been amended in response to Reviewer 2 Major comment 7. 166 11. Lines 461 – 467 have been amended in response to Reviewer 1 Major comment 1 167 and Reviewer 2 Major comment 7 168 12. Lines 475 – 567 have been amended in response to Reviewer 1 Major comment 3.1 169 and Reviewer 2 Major comment 2. 170 13. Lines 513-517 have been amended in Response to Reviewer 1 Major comment 3.2 171 14. Line 571 and Lines 577-579 has been amended in response to Reviewer 2 Major 172 comment 7 15. Lines 581 – 567 have been amended in response to Reviewer 1 Major comment 3.1 173 and Reviewer 2 Major comment 2. 174 175 176

**1 Manuscript with changes tracked**

[revised manuscript text omitted]